# Preparing for Practice: An Exploration of Health and Social Care Professionals’ Perceptions of Behaviour Change Education

**DOI:** 10.3390/bs15111523

**Published:** 2025-11-08

**Authors:** Hayley Breare, Chloe Maxwell-Smith, Deborah A. Kerr, Barbara A. Mullan

**Affiliations:** 1School of Population Health, Faculty of Health Sciences, Curtin University, Bentley, WA 6102, Australia; hayley.breare@postgrad.curtin.edu.au (H.B.); chloe.maxwell-smith@curtin.edu.au (C.M.-S.); d.kerr@curtin.edu.au (D.A.K.); 2Behavioural Science and Health Research Group, Curtin University, Bentley, WA 6102, Australia; 3Enable Institute, Faculty of Health Sciences, Curtin University, Bentley, WA 6102, Australia; 4Curtin Medical Research Institute, Faculty of Health Sciences, Curtin University, Bentley, WA 6102, Australia

**Keywords:** behaviour change, health and social care, tertiary education, behavioural medicine, health professionals

## Abstract

Health and social care professionals are important for fostering behaviour change to improve population health. Behaviour change education is varied across university curricula, impacting practitioner preparedness to promote engagement in health behaviours. This study examined health and social care professionals’ perceptions of behaviour change education and training in their university course and the factors influencing their preparedness to engage in behaviour change conversations, guided by the Theoretical Domains Framework (TDF). Australian health and social care professionals (N = 153, *M*_age_ = 33.4, *SD* = 10.5) were surveyed on their perceptions of behaviour change training, knowledge, confidence, and six TDF domains. Sixty-one percent of participants reported that communication skills were highly integrated (‘a lot’ to ‘a great deal’) throughout their course, compared to behaviour change techniques (45.8%), behaviour change theories (45.8%), and counselling therapies (39.9%). Mental health/social care professionals differed significantly from primary care and allied health professionals in skills (*p* < 0.05) and beliefs about capabilities (*p* < 0.05 primary care only). Findings demonstrated strong professional identity and intentions for behaviour change but lower confidence in their own capability to deliver behaviour change interventions. University curricula should expand behaviour change content beyond current communication skills training, using discipline-specific approaches for improved graduate preparedness.

## 1. Introduction

Behaviour change is fundamental to improving health, wellbeing, and reducing the burden of disease worldwide ([44]). In Australia, 85% of the burden of disease is attributed to non-communicable diseases (Australian Institute of Healthand Welfare, [10]). The prevention and management of non-communicable disease require addressing modifiable behavioural risk factors, such as insufficient physical activity, poor diet, and harmful use of tobacco and alcohol ([58]). Many Australians do not meet the health recommendations for health-enhancing behaviours, with 78% of adults insufficiently engaging in physical activities and 96% not meeting the guidelines for fruit and vegetable consumption ([11], [14]). Concurrently, the increasing prevalence of modifiable social determinants of poor health such as unemployment, limited housing, and reduced access to healthcare ([15]) necessitates individual- and community-based interventions. Researchers have made calls to action for the integration of behaviour change techniques in health and social care contexts, including emphasising the importance of behavioural medicine in pharmacy, mental health services, and oral health practices ([4]; [26]; [34]). Health and social care professionals have a valuable role in delivering evidence-based behaviour change strategies to support patients across clinical and community settings ([32]).

In Australia, health and social care professionals work within primary care and allied health where their roles involve the provision of specialised support for individuals and communities to improve health and wellbeing ([6]; [13]); this includes medical practitioners, psychologists, physiotherapists, social workers, and pharmacists. Australia’s National Preventative Health Strategy (2021–2030) emphasises the role of health professionals in supporting patients to adopt healthier behaviours to improve public health ([23]). Increasingly, health and social care roles will be expected to deliver behaviour change interventions, also known as “behaviour change conversations” ([19]; [23]; [27]). These conversations involve the provision of behaviour change advice or support as part of preventative practice to improve population health outcomes. Provision of such support requires comprehensive communications skills (e.g., motivational interviewing) and an understanding of theoretical approaches and behaviour change techniques for effective intervention delivery ([48]; [55]). At times, interventions may also require health communication skills that extend beyond the individual to address community and organisational contexts ([43]). Despite education being a key component in patient care, it is often not sufficient for behaviour change ([3]). Interventions should not rely solely on patient education but rather should incorporate active behavioural strategies, such as behaviour change techniques ([39]; [49]). Behaviour change techniques are described as the building blocks of interventions that are targeted and intended to change behaviour ([39]). Across health behaviours, specific behaviour change techniques have shown to be effective for improving health outcomes, such as goal-setting and self-monitoring techniques for healthy eating and physical activity interventions ([38]; [50]), practical social support, and restructuring the environment which are recommended for pharmacists’ promotion of medication adherence ([34]). Interventions designed using behaviour change theory and techniques, when delivered by health professionals, have shown potential to improve patient outcomes ([46]). However, despite behaviour change techniques being important to optimising patient outcomes, health and social care professionals may not be equipped with behaviour change competencies crucial to foster effective health behaviour conversations, or they may not have the confidence to initiate them.

Curriculum reviews examining behaviour change education and training for health professionals in Australia have shown differences in content across and within health and social care courses, often lacking explicit teaching of behaviour change techniques and theory ([16]; [31]; [47]). For example, a curriculum review of Australian pharmacy courses showed limited reference to terms related to behaviour change in unit outlines, with educators describing behaviour change as a “nice to have” as opposed to a need ([16]). Even in placement opportunities where students have to translate knowledge into practice, opportunities to practice behaviour change skills are limited ([21]). Training could better support health and social care professionals by fostering their confidence, competence, knowledge, intentions, and use of behaviour change techniques ([19]; [27]; [36]; [42]). However, individual and organisational barriers limit the use of opportunistic behaviour change conversations; these barriers include lack of time, workload pressures, concerns about potential outcomes of their interaction with patients (beliefs about consequences), low practitioner confidence, and a need for further training ([29], [30]; [37]).

The disconnect between behaviour change education and barriers in practice highlights the need to understand how university education prepares health and social care professionals for real-world application of behaviour change skills. The Theoretical Domains Framework (TDF) is a comprehensive framework for understanding the factors that influence behaviours such as health professionals’ behaviour, which has been used in healthcare settings ([5]). The TDF can provide a structure for factors influencing the delivery of behaviour change interventions, thereby informing educational gaps across health and social care settings. The aim of the study was (1) to understand how health and social care professionals perceive university-based behaviour change training and (2) to explore, drawing on the TDF, the differences across factors that contribute to professionals’ preparedness to deliver behaviour change interventions.

## 2. Materials and Methods

A cross-sectional survey was developed to examine factors associated with health behaviour change education among health and social care professionals. Participants completed measures related to their confidence, competence, perceptions of their university course, and key domains from the TDF ([5]). Survey items are shown in Appendix A. Recruitment occurred in two phases, due to a low initial response rate. The first phase targeted recent graduates (<2 years post-graduation) from pharmacy and nutrition and dietetic programmes through social media, word of mouth, snowballing, and professional organisations. Upon completion of the survey, participants were invited to enter a prize draw to win one of three $50 vouchers (for each profession). Due to insufficient recruitment from these platforms, phase 2 expanded the eligibility criteria to include health and social care professionals through Prolific, an online paid recruitment platform. Participants were screened to determine if they were eligible and then were invited to complete the main survey. Prolific participants received £0.30 for completing the screener, with eligible participants receiving an additional £2.25 (more than minimum wage) upon completion of the main survey. Ethics approval was obtained from Curtin University’s Human Research Ethics Committee (HREC 2023-0026).

### 2.1. Participants

One hundred and fifty-three participants completed the survey. In Phase 1, participants had to have completed an Australian accredited university course in the last 2 years (i.e., in pharmacy or nutrition and dietetics) and to be currently qualified or working as a provisional in their profession. For Phase 2, participants were required to reside in and have completed their qualification in Australia, and to be currently working in a patient-facing health or social care role (e.g., social workers, support workers), where they were seeing or treating patients (i.e., face to face, telehealth, or both).

### 2.2. Measures

A definition and examples of health behaviour change were provided to participants at the start of the survey to ensure consistent interpretation of the term. Participants provided information on their age, gender, profession, year of registration, role, university, location, education level, and service delivery mode (e.g., telehealth). Items related to course characteristics, application, and professional development were developed for the purpose of this study. A pragmatic approach was used to address the research question, with item development being reviewed by an experienced Advanced Accredited Practising dietitian for Phase 1 of the study (nutrition and dietetics survey), and the wording of items was subsequently adapted in Phase 2 to reflect health and social care roles more broadly.

#### 2.2.1. Course Characteristics

Three questions were used to assess how health professionals perceived behaviour change education and training within their course. The first two questions were regarding the amount of focus the course had on health behaviour change and its integration in the course (i.e., building your skills across unit/topics and year level). This included the amount of behaviour change theories or models, counselling therapies, communication skills, behaviour change techniques, and using interventions designed to change behaviour beyond individual settings. The third question was regarding how much the course included behaviour change training in lectures, workshops and tutorials, assessments, and placements. All questions were rated on a 5-point Likert scale ranging from ‘None at all’ (0) to ‘A great deal’ (4). Higher scores indicated greater focus, integration, or inclusion in the course.

#### 2.2.2. Course Satisfaction

Course satisfaction was assessed using a single item that asked participants “Thinking about your course, how satisfied are you with how the components of the course have prepared you to deliver health behaviour change in your current/future role?” Participants rated this item on a 5-point Likert scale ranging from ‘Extremely dissatisfied’ (0) to ‘Extremely satisfied’ (4). Higher scores indicated a higher satisfaction with their course.

#### 2.2.3. Knowledge

Perceived knowledge was measured using single items for each of the following skills related to behaviour change. This included behaviour change theories or models, counselling therapies, communication skills, behaviour change techniques, and using interventions designed to change behaviour beyond individual settings. Participants were asked to rate their knowledge of health behaviour change following their course on a slider scale, ranging from no understanding (0) to full understanding (7). Scores were averaged to generate an overall knowledge score. Higher scores indicated higher perceived knowledge. This scale has been used in other studies on behaviour change among health professionals ([42]). The measure demonstrated good internal consistency in the current study (α = 0.87).

#### 2.2.4. Confidence, Perceived Importance, and Usefulness

Three items were used to assess participants’ confidence, perceived importance, and usefulness of engaging in behaviour change conversations with patients. This measure was adapted from a previous study conducted with health professionals in health behaviour change conversations ([27]). An example item includes, “How confident do you feel about supporting patients to make behaviour changes?” Items were rated on a 5-point Likert scale ranging from ‘Not at all’ (0) to ‘Extremely’ (4). Higher scores indicated higher confidence, perceived importance, and usefulness.

#### 2.2.5. Theoretical Domains Framework

Six TDF domains were selected to assess potential factors that contribute to health professionals’ preparedness to engage in behaviour change conversations in practice. These domains focus on individual-level determinants of behaviour, which contribute to health and social care professionals’ preparedness. Contextual or organisational influences were excluded to focus on factors amenable to individual-level interventions. These domains were guided by [28] ([28]) and [5] ([5]) which have been validated for use in health professional behaviours. Twelve items were included across six domains: skills (competence and practice), professional identity (role perceptions), beliefs about capabilities (self-efficacy), beliefs about consequences (anticipated outcomes), intentions and automaticity. Items were rated on a 7-point Likert scale ranging from ‘Strongly disagree’ (1) to ‘Strongly agree’ (7). One item was reversed scored. Examples of items include “Having behaviour change conversations with patients is part of my work as a health professional” and “I have been trained on how to have behaviour change conversations with patients.” Participants were instructed to reflect specifically on the knowledge and skills gained through their university course, to ensure responses reflected their formal education rather than any professional development or external training. Domain scores were calculated by averaging the items from each domain. Higher scores indicated positive perceptions of behavioural determinants and greater preparedness to implement behaviour change conversations in practice.

#### 2.2.6. Behaviour

Behaviour was assessed using a single item that asked, “In a typical week, how often do you have behaviour change conversations with patients or clients (e.g., about diet, physical activity, smoking, alcohol, or medication adherence)?” Participants rated this item on a 5-point Likert scale ranging from ‘None of the time’ (1) to ‘All the time’ (5). Higher scores indicated greater frequency of behaviour change conversations.

#### 2.2.7. Course Application

Perceived applicability of health behaviour change knowledge and skills was assessed using two single items that asked, “After completing your course, how applicable do you feel that the health behaviour change knowledge and skills acquired during your course are to your current or future professional role?” And “Following your course, how much do you feel you have (or had to) upskill your health behaviour change skills to work with patients?” Both items were rated on a 5-point Likert Scale with responses ranging from ‘Not applicable at all/Not at all’ (0) to ‘Extremely applicable/A great deal’ (4). Higher scores on items indicated greater perceived relevance of the training to professional practice. Higher scores on the second item indicated greater perceived need for additional upskilling beyond the course.

#### 2.2.8. Continuous Professional Development

The likelihood of seeking further education and training in behaviour change techniques was assessed using a single item, “How likely will you be to seek professional development in behaviour change techniques and approaches now your course has finished?” Participants rated this item on a 6-point Likert scale ranging from ‘Extremely unlikely’ (1) to ‘Extremely likely’ (5), with an option to indicate if they had already completed professional development in this area (6).

### 2.3. Data Analysis

In Phase 1, data was screened for duplicates (*n* = 3), missing data (>50% of survey items or missing demographic information; *n* = 22), and ineligibility (*n* = 1), leaving a sample of 53 participants. In Phase 2, 201 participants were screened via Prolific, with 144 deemed eligible and invited to complete the main study. The survey was further screened for ineligibility (*n* = 7) and incomplete data (*n* = 3), leaving 100 participants. Data from the two phases were merged for analysis in IBM SPSS Statistics (Version 30), resulting in a final sample of 153 participants. Descriptive statistics, including means or median, standard deviations, ranges, and frequencies were computed. Assumption testing indicated that distributions of variables were approximately normal, with minor deviations in skewness and kurtosis, which were within acceptable limits, and given the sample size, were not considered problematic ([54]). Outliers in TDF domains were identified but were not deemed problematic and were retained in the analyses. A Mann–Whitney U was used to determine if there were differences between the samples from each phase. This revealed a significant difference in age between Phase 1 (*Md* = 25, IQR = 24.00–28.00, *n* = 53) and Phase 2 (*Md* = 34, IQR = 27.25–44.00, *n* = 100), U = 1307.50. z = −5.15, *p* < 0.001, *r* = 0.42, with Phase 2 participants being significantly older. There were no other significant differences between samples. Group comparisons were made based on health and social care roles, and, where appropriate, non-parametric tests such as the Mann–Whitney U were used to account for unequal group sizes and variances.

## 3. Results

### 3.1. Demographic Characteristics

Participants were health and social care professionals who had a mean age of 33.4 (*SD* = 10.5, age range = 20–65 years). Participants were mainly women (81%), with almost half of participants having a Master’s degree (45.8%). Many participants were early-career health or social care professionals (66.7%), with accreditation years spanning from 1985 to 2025. Participants were categorised based on their primary setting and scope of practice into three groups: (1) primary care which included hospital physicians, general practitioners, nurses, pharmacists, dentists, radiographers, and first responders; (2) allied health which comprised dietitians, rehabilitation and functional therapists, podiatrists, speech pathologists, radiation specialists, optometrists, and cardiac physiologists; and (3) mental health/social care which comprised social workers, disability support workers, psychologists, mental health professionals, and counsellors. Demographics for the full sample are shown in Table 1.

### 3.2. Perceptions of Education and Training

Only 18.3% of participants perceived the behaviour change knowledge and skills that they had acquired during their course as inapplicable to their current role, while half of the participants (50.1%) felt that they needed to upskill in behaviour change competencies. Participants (*n* = 100) reported engaging in behaviour change conversations most or all the time (44%), some of the time (39%), or were rarely or not engaging in behaviour change conversations at all (17%). In terms of seeking professional development, most participants indicated that they were ‘likely’ or ‘very likely’ (70%) to pursue professional development opportunities in behaviour change techniques, with only 12% indicating that they had already completed professional development.

### 3.3. Course Characteristics[note 1]

Overall, participants reported varying perceptions of the amount of behaviour change content in their health and social care course. Participants reported communication skills as the most emphasised component in their course, with approximately two-thirds of participants (65.0%) indicating a substantial focus (‘a lot’ to ‘a great deal’), while only 10.5% reported minimal emphasis on communication (none to a little). In contrast, the focus on behaviour change interventions beyond individual settings appeared to receive the least amount of focus, with 44.4% of participants reporting minimal focus, compared to 25.4% indicating substantial focus. The remainder of the behaviour change concepts showed more balance, with participants reporting either a moderate or substantial amount of content on behaviour change theories (moderate: 40.5%; substantial: 26.1%), counselling therapies (moderate: 30.0%; substantial: 35.9%), and behaviour change techniques (moderate: 32.0%; substantial: 37.3%). Integration was defined as building behaviour change skills across units/topics[note 2] and across year levels. Participants indicated that there was moderate (25.5%) or substantial (54.9%) integration of training on communication skills within their course. Comparatively, participants perceived little to no integration of behaviour change theories (45.8%), content on interventions beyond individual settings (45.1%), counselling therapies (39.9%), or behaviour change techniques (45.8%) within their course.

### 3.4. Behaviour Change Content

Behaviour change education and training varied across health and social care professionals. Almost a third of participants reported little to no content on behaviour change provided across placements (31.4%), assessments, (30.7%), lectures (24.2%), and workshops and tutorials (26.8%). However, moderate to substantial inclusion of behaviour change content was reported for lectures (moderate 39.9%; substantial 36.9%), compared to tutorials and workshops (moderate 30.1%; substantial 43.1%), assessments (moderate 30.1%; 39.2%), and placements (moderate 28.8%; substantial 39.8%).

### 3.5. Satisfaction with Course

Overall, most participants were satisfied with their course (69.9%), including satisfaction with available opportunities to practise behaviour change skills (e.g., placements, practical exams; 56.2%). Over half were satisfied with the amount of behaviour change content (56.9%), teaching quality (57.5%), and the skills they acquired (56.2%). Descriptive statistics for the focus, integration, content, and satisfaction of course components are presented in Table 2.

### 3.6. Perceived Knowledge, Confidence, Importance, and Usefulness

Of the areas measured, participants rated their knowledge of communication skills as the highest (*M* = 5.54, *SD* = 1.36), followed by counselling therapies (*M* = 4.06, *SD* = 1.95), and behaviour change techniques (*M* = 4.22, *SD* = 1.67). Lower knowledge was reported for behaviour change theories and models (*M* = 3.90, *SD* = 1.70) and interventions beyond individual settings (*M* = 3.88, *SD* = 1.80). Across all knowledge domains, participants demonstrated a moderate knowledge score (*M* = 4.32, *SD* = 1.38). In addition, 82.4% of participants were ‘moderately’ to ‘extremely’ confident about supporting patients to make changes to their behaviour, while 96.7% indicated that supporting patients to make changes to their behaviour was moderately to extremely important to them. Similarly, 88.9% perceived behaviour change conversations with patients as useful.

### 3.7. Theoretical Domains Framework

All TDF domains demonstrated slight deviations from normality (Shapiro–Wilk *p* < 0.001) with negative skewness, indicating scores clustered toward the higher end of the scale. Therefore, medians with interquartile ranges are reported. Overall, participants’ perceptions of behaviour change practice as part of their professional identity was the highest (*Md* = 6.50, IQR = 5.50–7.00), followed by positive beliefs about the consequences of behaviour change initiatives (*Md* = 6.00, IQR = 5.50–6.50), and intentions to implement behaviour change initiatives (*Md* = 6.00, IQR = 5.00–7.00). Participants perceived their skills (*Md* = 5.33, IQR = 4.67–6.00), beliefs about their capabilities (*Md* = 5.00, IQR = 4.00–5.50), and automaticity (*Md* = 5.00, IQR = 4.00–6.00) for behaviour change conversations to be moderate.

### 3.8. Differences Between Groups

Kruskal–Wallis Tests revealed a statistically significant difference in skills for behaviour change across the three groups (primary care, n = 56; allied health, n = 65, mental health/social care, (n = 32), χ^2^ (2, n = 153) = 10.43, *p* < 0.05). Post hoc comparisons using Bonferroni corrections showed that mental health/social care professionals (*Md* = 5.67, IQR = 5.00–6.67) had a significantly higher median score than primary care (*Md* = 5.17, IQR = 4.67–6.00, adjusted *p* = 0.03) and allied health (*Md* = 5.00, IQR = 4.33–6.00, adjusted *p* = 0.005). Similar results were found for beliefs about capability, (χ^2^ (2, n = 153) = 10.27, *p* < 0.05), with post hoc comparisons revealing a significant difference between mental health/social care professionals (*Md* = 5.50, IQR = 4.50–6.00) and primary care (*Md* = 4.25, IQR = 3.50–5.50, adjusted *p* = 0.004). However, there was no significant difference between mental health/social care professionals and allied health (*Md* = 5.00, IQR = 3.75–5.50) after adjustments for Bonferroni corrections(adjusted *p* = 0.07). No other significant differences were found between professional groups for the remaining TDF domains (professional identity, beliefs about consequences, intentions, and automaticity).

## 4. Discussion

This is the first exploratory study to examine Australian health and social care professionals’ perceptions of behaviour change education and training and the factors that contribute to their preparedness to engage in behaviour change conversations. Health and social care professionals displayed variability in behaviour change training during their university education. Over half of the participants reported that communication had a substantial focus and integration throughout their course, compared to behaviour change theories, counselling therapies, behaviour change techniques, and using interventions beyond individual settings. Overall, findings from the TDF domains were moderate to high across professional identity, intentions, skills, beliefs about capabilities, and automaticity of delivering behaviour change conversations. However, both primary care and allied health professionals rated their skills significantly lower than mental health/social care professionals, while primary care professionals also scored significantly lower on beliefs about capabilities.

Health and social care professionals consistently reported that communication skills received considerable attention in their training and that it was integrated throughout their course. Participants correspondingly rated their knowledge of communication skills as the highest, which was anticipated, given that communication skills are essential to support effective delivery of behaviour change interventions ([20]). Behaviour change content is primarily focused on communication and counselling skills ([16]; [47]). However, the language used to describe terms related to behaviour change theory and techniques may vary between disciplines ([22]; [57]). Findings from the current study suggest that behaviour change theories and techniques may be less prioritised within health and social care courses. This could be attributed to the ‘need-to-know’ principle, whereby course competencies focus on core content and skills, with limited available time or priorities allocated to behaviour change skills ([51]). For example, a review of the Australian National Competency Standards for pharmacists included only foundational behaviour change competencies but demonstrated gaps in competencies for high-intensity behaviour change interventions ([53]). This is also reflected in the Australian curricula where minimal reference to behaviour change in course content and learning outcomes was observed in pharmacy courses ([16]). Similar gaps may exist across other health and social care professions, though the extent to which behaviour change is embedded in competency standards and curricula likely varies by discipline and professional context. Alternatively, participants in the current study completed their degree between 1985 and 2025, which means findings may reflect differences not only across courses but also changes within university curricula and competency standards. For example, revisions of competency standards have been made for Australian dietitians to ensure they remain up to date with the current health workforce requirements and evidence-based standards ([1]). However, registration standards are regularly reviewed for specific health professions every five years and demonstrate variability in how curriculum changes are implemented across institutions ([35]). Tertiary institutions may benefit from leveraging existing curriculum review processes to specifically strengthen behaviour change integration across their courses and draw comparisons of cohorts trained before and after to determine the impact of the changes on behaviour change competencies.

In the current study, health and social care professionals viewed behaviour change as important and useful to their role. However, participants rated lower knowledge on behaviour change theories and interventions beyond individual settings compared to other behaviour change areas. Behaviour change theories and frameworks are derived from health psychology, where theories such as the COM-B model (Capability, Opportunity, Motivation and Behaviour model; [40]), Health Action Process Approach model ([52]), and the behaviour change technique taxonomy (v1) ([39]) are commonly used across behaviour change interventions ([24]; [41]). However, theoretical approaches may differ within health and social care courses. In addition, knowledge of interventions beyond individual settings may be more applicable to community-based professionals who implement population health programmes, health promotion initiatives, and group interventions, compared to professionals focused primarily on individual patient care ([12]; [56]). Tailored ongoing professional development opportunities should address role-specific needs, such as teaching individual- and community-based theoretical frameworks and behaviour change techniques, where applicable for clinical and community settings (e.g., hospital, private practice). Training interventions for health professionals exist and have shown promising results in improving professionals’ confidence, intentions, motivation, and implementation of behaviour change techniques ([17]; [36]; [42]). Educational and practical training can be effective at improving the delivery of behaviour change interventions and patient outcomes in short and medium time frames ([25]). However, professional development opportunities in behaviour change can be limited or difficult to access. For example, general practitioners and general practice nurses report a lack of availability of professional development opportunities on behaviour change ([18]). This suggests that despite evidence supporting the effectiveness of behaviour change training, systemic barriers may prevent accessibility of such training.

Professionals perceived themselves as well-prepared to engage in behaviour change conversations, based on our TDF findings. Mental health and social care professionals had higher perceived skills and beliefs about their capabilities compared to primary care (beliefs about capabilities only) and allied health. Differences among groups may be attributed to variation in primary role demands. For example, [29] ([29]) found that general practitioners recognised the importance of behaviour change interventions but experienced physical opportunity barriers such as time, workload pressures, and a lack of prioritisation of behaviour change in practice. Comparatively, in mental health and social care roles, psychological theories and techniques for behaviour change are foundational components of their professional competencies ([9]). In addition to differences in primary role demands, there may also be differences in how these roles embody behaviour change education and training opportunities in tertiary curricula, such as a variation in reliance on experiential opportunities, which are essential for developing and strengthening behaviour change competencies ([21]; [33]). Furthermore, as shown in the current study, few participants had completed continuous professional development in behaviour change techniques, highlighting that opportunities to strengthen and rehearse these skills post-graduation may be limited. While continuous professional development is a mandatory requirement for health and social care professionals to engage in ongoing learning post-graduation, the types of activities required differ across professions and are guided by individual regulatory bodies, which may not emphasise behaviour change as a core competency requirement ([8]; [45]).

This exploratory study provides an overview of health and social care professionals’ perceptions of behaviour change education and training and their preparedness to engage in behaviour change conversations. Our sample size was relatively small due to recruitment difficulties in Prolific’s participant pool and low participant engagement in Phase 1. This may have resulted in a limited representation of health and social care professionals in Australia. This also limited our ability to draw subgroup comparisons within primary care, allied health, and mental health/social care roles. Research facilitating stratification across disciplines and subgroups would allow such comparisons. As we had an overrepresentation of dietitians in our sample, quota sampling through professional regulatory bodies or evaluations conducted as part of university curriculum reviews would ensure proportional representation across and within professions. In addition, participants who have not recently completed their course may have also presented potential recall bias ([2]) or variability in course requirements and content. As difficulties in recruitment for early graduates impacted the generalisability of findings regarding current course content and delivery, future research should partner directly with universities to systematically sample recent graduates (within 12 months of completion) as part of routine curriculum and competency review processes. Within this process, future research should incorporate qualitative responses, such as interviews or open-text questions, to capture richer insights into recent graduates’ experiences of their behaviour change training as part of their tertiary education.

Addressing the prevalence of modifiable disease risk is urgently needed to reduce the burden of disease from chronic conditions ([7]). The capacity of health and social care professionals to support individuals in initiating and sustaining health behaviour change is integral to chronic conditions prevention and management. To our knowledge, this is the first study to examine health and social care professionals’ perceptions of behaviour change training within Australian university education and their preparedness for practice, offering insights to guide educators in developing confident and competent practitioners. Our study identified gaps in behaviour change training that could be addressed through teaching approaches on theoretical models, counselling therapies, and integration of behaviour change techniques. TDF domain scores revealed that fostering skills and beliefs about capabilities were important factors that may be targeted in Australian tertiary education to build practitioner confidence. The findings underscore the need to embed behaviour change training within Australian health and social care programmes, which may strengthen professionals’ competencies to deliver effective evidence-based behaviour change interventions in practice, and ultimately improve patient outcomes and preventative care. As such, this may promote the integration of behaviour change techniques into routine clinical care. Future research should explore additional TDF domains to better account for organisational and external influences, offering insights to guide educators in strengthening behaviour change curricula for Australian health and social care professionals.

## Figures and Tables

**Table 1 behavsci-15-01523-t001:** Health and social care professional characteristics (N = 153).

	n	%
Gender		
Woman	124	81%
Man	27	17.6%
Non-binary	1	0.7%
Prefer not to say	1	0.7%
Location		
Victoria	48	31.4%
New South Wales	39	25.5%
Queensland	27	17.6%
Western Australia	20	13.1%
South Australia	10	6.5%
Australian Capital Territory	9	5.9%
Tasmania	0	0%
Northern Territory	0	0%
Profession		
Dietitian	46	30.1%
Medical professionals: General practitioners, physicians, nurses, radiographers, radiation specialists, cardiac physiologists	41	26.8%
Mental health professionals: Psychologists, Psychiatry assistants, counsellors	23	15%
Pharmacists	12	7.8%
Rehabilitation and functional therapist: Physiotherapy, exercise physiology and occupational therapy	10	6.5%
Social care: Social worker and disability support, alcohol and drug clinician	9	5.9%
Dentistry	5	3.3%
Specialist health professionals: Optometrist, speech pathologist and podiatrist	5	3.3%
Frontline workers: Paramedic, patient transport	2	1.3%
Role Type ^1^		
Private Practice	57	37.2%
Community	46	30.1%
Hospital—Acute/Subacute	33	21.5%
Public health	23	15.0%
Hospital—Clinical	20	13.1%
Other ^2^	19	12.4%
Hospital—Pharmacy	4	2.6%
Mental health	16	10.5%
Aged Care	14	9.15%
Research	13	8.5%
Disability	4	2.6%
Years since graduation		
Early career 2020–2025	102	66.7%
Mid-career 2009–2019	39	25.5%
Late career 1985–2008	12	7.8%
Frequency of behaviour in a typical week (*n* = 100)		
None of the time	2	2%
Rarely	15	15%
Some of the time	39	39%
Most of the time	30	30%
All the time	14	14%

Note. ^1^ Participants could choose more than one role type. ^2^ Other includes emergency response, generalist rural, sports medicine, food industry, locum pharmacy, occupational health, and general practice.

**Table 2 behavsci-15-01523-t002:** Perceptions on behaviour change training within health and social care courses (N = 153).

Course Characteristics	*M*	*SD*
Course focus on behaviour change		
Behaviour change theories or models and how to implement them	1.94	1.02
Counselling therapies	2.07	1.17
Communication skills	2.82	0.99
Behaviour change techniques	2.16	1.05
Using interventions designed to change behaviour beyond individual settings	1.80	1.11
Integration (i.e., building skills across units/topics and year levels) of behaviour change		
Behaviour change theories or models and how to implement them	1.75	1.05
Counselling therapies	1.89	1.16
Communication skills	2.60	1.10
Behaviour change techniques	1.96	1.02
Using interventions designed to change behaviour beyond individual settings	1.80	1.13
Behaviour change content		
Lectures (e.g., theory)	2.19	0.97
Workshops and tutorials (e.g., practical opportunities and role plays)	2.25	1.07
Assessments (e.g., practical assessments, OSCEs ^1^, written, oral presentations, group work)	2.18	1.18
Placements (e.g., community, hospital)	2.14	1.32
Satisfaction		
Teaching quality of health behaviour change	2.41	1.02
Skills learned to change behaviour (e.g., counselling strategies and techniques)	2.42	1.13
Opportunities to practice skills in health behaviour change (e.g., OSCEs, placements)	2.37	1.17
Overall course	2.70	0.86

Note. ^1^ Objective Structured Clinical Examinations (OSCEs) include a systematic, station-based format where students are assessed on their clinical competencies such as counselling and clinical reasoning skills using patient simulations.

## Data Availability

The data are not publicly available due to ethical restrictions.

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
