# Peer review of "Preparing for Practice: An Exploration of Health and Social Care Professionals’ Perceptions of Behaviour Change Education"

_behavsci, 2025, doi:10.3390/bs15111523_

Round 1
Reviewer 1 Report
Comments and Suggestions for Authors
Thank you for the opportunity to read this really interesting paper. I would agree that training and education are an important conduit to establishing the behaviour change skills and expertise in the workforce that can ultimately assist in changing population health behaviours. It is important to see research such as yours which seeks to establish perceptions of that training.
Some specific and general comments below:
This paper would have been enhanced by accompanying it with at least a brief audit of existing provision by exploring the curricula of some institutions. This would provide a useful context in which to consider the perceptions of individuals about the training/education they had received.
Similarly the amount of training does not necessarily equate to where gaps might be, which might be around recall, or retention but might also be as you have indicated in the introduction, that it has not sufficiently prepared practitioners for delivering in situ - perhaps because there was limited role playing or other interactive type of training. Your question I have had to upskill since taking the course, helps with establishing perceptions around need.
Abstract:
You note in your first sentence of your abstract (and later in the body of your manuscript) that health care professionals behaviour change is essential. While I would tend to agree that it is very important, we do know that behaviour change can take place in other ways and not through pivotal conversations with health care professionals. I would temper this from necessary to important.
Line 15: Be certain to include both education and training - first is knowledge and the second is skills based so important to include both if this is what you are looking for.
line 27: full stop missing
Line 44: delete first the in the sentence (before emphasising).
Line 52: take away full stop
Line 135: behaviour change education AND training
Line 171: not clear why you selected only 6 TDF domains when I could imagine that others such as environmental context and resources and social support may have also been very important to consider. Nevertheless, it is important to explain your rationale for the limitation here.
Line 181: Your question statement - I have been trained in ... but it does not specify that this training took place while at university. Is there not a risk that people may be referring to other training?
Method: Overall it would have been helpful to see the full questionnaire
Line 226: You have included behaviour change knowledge as demographic information. Interesting choice. I feel it would be better placed below participant characteristics.
Line 239: the demographics do not always add up to 153 and it would be useful to see why this is (see gender and year since graduation).
Results: It would be good to see a comparison of perceptions of training /education of those who trained prior to say 2010 or to identify a point where curricula changed according to policy in your context (i.e. you note the significance of a policy in 2022). This would help to establish if the views about behaviour change education and training are from current programmes or from programmes delivered many years ago before behaviour change became more common in health and social care training programmes.
Discussion is interesting and summarises results well. Indeed you mention some of my notes from above i.e. the challenge of collating all participants in one group rather than distinguishing those who trained many years ago. I feel the discussion might have benefited from noting the availability of cpd training, and whether that figures prominently in health and social workforce training requirements.
Line 256: not a full sentence
Again, many thanks for the opportunity to read this. It is so valuable to see this type of work.

Reviewer 2 Report
Comments and Suggestions for Authors
This is a useful, well-written and presented paper which addresses perceptions of professional behaviour change training in Australian health and social care professionals. It makes a coherent argument for the necessity of such training, and for training which is high quality in terms of being theory and skills based.
I found it interesting, although not really ground-breaking in terms of originality. I think it is challenging to deliver effective behaviour change training which has a lasting impact on practice, and it is even more difficult to demonstrate the impact of such training on population health. I would like to have been more convinced by the authors of the importance of the issues discussed, and relevance for specific professional groups or individuals, illustrated with more grounding in concrete examples from the Australian context.
Overall I found some components of the paper lacked the specificity which would make it very helpful to future educators to identify the key components of such training - to have short term and longer term impact. It would have been useful to know which specific aspects of the training (eg which BC techniques and skills) had most impact for which professionals, with which patient groups. (including some qualitative data or quotes reflecting these experiences in context would be very helpful for the non -Australian reader).
It seemed that most participants were generally very positive about the training and its utility - what makes me wonder about potential biases in recruitment, sampling (this was a relatively small sample which over-represented dieticians as a professional group), retrospective recall, and reporting? Although these are discussed it would be good to see more weight given to a critical discussion to identify solutions to these limitations in more depth.
Round 2
Reviewer 1 Report
Comments and Suggestions for Authors
I have now reviewed the corrections by the authors and am happy to recommend acceptance of this manuscript for publication.
Author Response
Thank you for your feedback and for recommending acceptance of our manuscript. We greatly appreciate the time and effort you have dedicated to reviewing our work throughout this process.